# Insight into the Molecular Basis Underlying Chromothripsis

**DOI:** 10.3390/ijms23063318

**Published:** 2022-03-19

**Authors:** Katarzyna Ostapińska, Borys Styka, Monika Lejman

**Affiliations:** 1Student Scientific Society of Laboratory of Genetic Diagnostics, Medical University of Lublin, A. Raclawickie 1, 20-059 Lublin, Poland; katarzyna1ostapinska@gmail.com; 2Borys Styka and Monika Lejman Laboratory of Genetic Diagnostics, Medical University of Lublin, A. Raclawickie 1, 20-059 Lublin, Poland; borys.styka@wp.pl

**Keywords:** chromoanagenesis, chromothripsis, chromoanasynthesis, cancer, TP53, gene alterations

## Abstract

Chromoanagenesis constitutes a group of events that arise from single cellular events during early development. This particular class of complex rearrangements is a newfound occurrence that may lead to chaotic and complex genomic realignments. By that, chromoanagenesis is thought to be a crucial factor regarding macroevolution of the genome, and consequently is affecting the karyotype revolution together with genomic plasticity. One of chromoanagenesis-type of events is chromothripsis. It is characterised by the breakage of the chromosomal structure and its reassembling in random order and orientation which results in the establishment of derivative forms of chromosomes. Molecular mechanisms that underlie this phenomenon are mostly related to chromosomal sequestration throughout the micronuclei formation process. Chromothripsis is linked both to congenital and cancer diseases, moreover, it might be detected in subjects characterised by a normal phenotype. Chromothripsis, as well as the other chromoanagenetic variations, may be confined to one or more chromosomes, which makes up a non-uniform variety of karyotypes among chromothriptic patients. The detection of chromothripsis is enabled via tools like microarray-based comparative genomic hybridisation, next generation sequencing or authorial protocols aimed for the recognition of structural variations.

## 1. Introduction

Significant progress within the field of molecular diagnostics in medicine has led to the identification and distinguishing of certain cancer subtypes and the structural aberrations underlying them. One of the possible explanations of the oncogenesis process is the dogma of macroevolution. It considerably defies the conventional theory of cancer progression driven by mutational acquisition. Over the last decade, new types of massive and complex chromosomal rearrangements based on the chaotic shattering and restructuring of chromosomes have been identified in cancer cells, patients with congenital diseases, as well as in healthy individuals. These unanticipated phenomena are named chromothripsis, chromoanasynthesis and chromoplexy, and are grouped under the term of chromoanagenesis (for chromosome rebirth) [1] (Figure 1). Despite their complexity, these specific alterations typically occur during a single cellular event. Furthermore, the molecular mechanisms characteristic for chromothripsis, chromoanasynthesis and chromoplexy differ substantially. Recent investigational data enabled us to describe and establish certain hallmarks of chromoanagenesis-derived alterations. It has also enabled one to consider possible mechanisms underlying these phenomena [2,3].

Chromoanasynthesis (synthesis of new chromosomes) was first described back in 2011 by Liu et al. and was referred to as a complex genomic rearrangement involving multiple breakpoint junctions [4]. Chromoanasynthesis itself is arising within the replication process that involves serial fork stalling and template switching (FoSTeS) or microhomology-mediated break-induced replication (MMBIR) mechanisms. The phenomenon of chromoanasynthesis defines the occurrence of localised multiple copy-number changes, particularly region-focused duplication and triplication, along with short stretches of micro-homologies at the breakpoint junctions. The aforementioned events are both the hallmarks of replication-based mechanism. A high incidence of marker chromosomes has been reported in patients with chromanasynthesis derived disorders. The main feature differentiating chromoanasynthesis from chromothripsis or chromoplexy is the fact that chromoanasynthesis-driven changes typically occur within a single chromosome and the aforementioned copy number variants tend to cluster in a new position within the chromosome [5]. Phenotypically, patients with chromoanasynthesis are displaying dysmorphic features, autism spectrum disorders, intellectual disabilities and developmental delay, however, they can also represent a mild phenotype [6,7,8].

Chromoplexy, as opposed to other chromoanagenetic events, does not encompass the DNA amplification along with replication process or presence of copy number variations (CNVs). Chromoplexy is characterised by the interdependent occurrence of multiple inter- and intra-chromosomal translocations and deletions, including up to eight chromosomes and presumably occurring simultaneously [9]. Chromoplexy was first reported by Baca et al. in prostate cancer. Statistical analysis performed by Baca et al. has shown multiple structural alterations occurring in a coordinated fashion, with unique patterns specific to the prostate cancer subtype. Rearrangements, numbering from 3 to over 40 and involving up to 7 chromosomes in a single chain, occur in the vast majority of the prostate cancers studied. In contrast to chromothripsis, chromosomal rearrangements present little or no copy number alterations [10]. The chromoplexy phenomenon supports the model for the punctuated tumour evolution characteristic for various subtypes of malignancies. Molecular mechanisms underlying the chromoplexy process are restricted to deletions and chained chromosomal rearrangements.

The mechanism of chromothripsis was characterised first in 2011 as a result of a single cellular event in which one or several chromosomes’ segments were broken in pieces and reassembled in a random order and orientation with the purpose to form complex derivative chromosomes. A phenomenon was termed chromothripsis (Greek, chromos for chromosome; thripsis, shattering into pieces). Being present in both cancer and congenital diseases, chromothripsis is not distinctive to merely cancerogenesis. Regarding cancer, chromothripsis is associated with its aggressive types and poor prognosis; therefore, it is at utmost importance to understand the mechanisms underlying this particular phenomenon [11]. Criteria for chromothripsis in tumour genomes were described by Korbel and Cambell in 2013. Chromothripsis is distinguished by (a) clustered breakpoints; (b) the oscillation of copy number states between one (deleted fragments with loss of heterozygosity) and two (with maintained heterozygosity); (c) rearrangements affecting a single haplotype (one of two homologous chromosomes); (d) random order and orientation of the DNA fragments within the derivative chromosomes; and (e) the ability to “walk” through the derivative chromosome by joining the breakpoints if all the breakpoints are available [12].

This review focuses on the molecular basis of the chromothripsis phenomenon and its significance in cancer and other disease development.

## 2. Chromothripsis (cth)

Chromothripsis defines as to the localised shattering and reshuffling of one or a few chromosome segments during a one-step catastrophic event, with the incomplete repair of double-strand breaks (DSBs) through non-homologous end-joining (NHEJ). After chromothripsis, a rearranged chromosome, deleted fragments and extra-chromosomal material (double minute) may be formed. Chromothripsis is characterised by multiple clustered chromosome breakpoints, a change in the low DNA copy number and by preservation of heterozygosity in the rearranged segments [13].

According to the International System for Human Cytogenomic Nomenclature, chromothripsis has been defined as the occurrence of complex patterns of gene copy number alterations along the full length of the chromosome or within a particular chromosome segment [14]. The hallmark of the chromothripsis process is that, in addition to the general restructurisation, amplified regions are also observed. Chromothriptic cluster rearrangements may reach up to thousands and they occur in an utterly random manner. Chromothripsis identification relies on the abovementioned criteria, however, these features are not exhaustive, and since the area of chromothripsis research is still developing, the total profile of chromothripsis hallmarks remains a point of debate and further examination [13].

### 2.1. Mechanisms Underlying the Chromothripsis Process

As the research on the chromoanasynthesis-type of events is evolving, their molecular origin has often been called into question. Initially, the scientific literature has been suggesting exogenous factors that were supposed to severely impact the double-stranded DNA structure [15,16]. This arising theory has been undermined due to the fact that exogenous stress factors do not impact the genome selectively when chromothripsis may be restricted to certain arms of chromosomes, although it never applies to the entire genome [17]. Back in 2011, Tubio and Estwill suggested several explanations for the chromothripsis onset. Abortive apoptosis, telomere attrition, mitotic errors, premature chromosome condensation, p53 defect, or viral integration were considered as cellular processes driving chromothripsis [18].

Proceeding from the deregulation of the cell cycle, one of the possible mechanisms underlying the chromothripsis phenomenon is associated with breakage-fusion-bridge (BFB) cell cycles. These mechanisms consist of the repetitive breakage of chromosomes followed by a re-joining process. During cell division in a homeostatic environment, telomeres of somatic cells undergo the process of shortening. Shortening occurs physiologically in every one of the cell cycles in order for telomeres to reach the critical length and to proceed with the physiological fate of the cell—the programmed death (apoptosis). However, for malfunctioning cells that did not acquire functional genes essential for the cell cycle (i.e., *TP53* or *RB* genes), the course of cell arrest is significantly disordered and, as a result, it is not executed. Whilst the affected cells continue to multiply, their non-functional telomeres are successively shortening. This particular abnormality leads to the exhibition of pathologically altered telomeres and their, now, “unprotected” structure. In this situation, the cell cycle proceeds to go on further, although the whole process is not supported by preventive mechanisms that exercise protection over proper cellular division and subsequent cellular events. Regarding those events, the “unprotected” state of telomeres predisposes to their fusion (occurring in a random manner), leading to the formation of dicentric chromosomes. Such a structure is characterised by the presence of two centromeres on a sole chromosome. Later, during mitosis the chromatin bridges are formed and, due to the dicentric character of the chromosomes, the bridges are rupturing, which then drives the cell into the BFB cycles. During anaphase, centromeres of dicentric chromosome can obtain different directions in which they continue stretching. The dicentric chromosome is spread between the two daughter cells. Under such circumstances, the final segregation cannot be executed and the chromatin bridges are consequently arising, thus further driving the abnormal conditions for gene amplification, causing multiple inversions, and deletions arisen due to the cleavage activity of TREX1 (cytoplasmic 3′-exonuclease) which, in this process, is aimed at the destruction of chromatin bridges [19,20]. Errors within the cell cycle give rise to more complex mechanisms causing cth, such as micronuclei fragmentation or abortive apoptosis [3,21].

The fragmentation of chromosomes within the micronuclei is thought to be the most imminent and relevant pathway that may induce chromothripsis. A comprehensive view on DNA damage and the mechanisms occurring in the micronuclei remains a matter of consideration and further research. Albeit, the association between DNA damage is established and well documented. Currently, there are a few possible explanations on how the damage within the micronuclei is setting. Beginning at the replication process, a delayed replication within the micronuclei may interfere with further replication processes, hence promoting the occurrence of double-strand DNA breaks. Upon the cell’s entrance to the mitosis cycle, multiple repair-associated mechanisms will start and the repair machinery will take appropriate actions regarding the aforementioned DSBs. These mechanisms remain crucial in the aspect of chromothripsis occurrence, since the chaotic repair of massive aberrations in the DNA ought to be the initial point of chromothripsis. The secondary theory on micronuclei-derived chromothripsis involves the process of disruption within the micronuclear envelope. A breakage of the envelope’s continuity leads to the disruption in its preserved microenvironment, leading to the exposure of its content to the external components of the cytoplasm. This causes DNA shattering, followed by its random reincorporation in the main nucleus [2,17,22].

Due to the existing evidence on abortive apoptosis being a trigger for cancer-specific chromosomal rearrangements, the aforementioned Tubio and Estwill took this approach into consideration [18]. The process is supposed to begin within the initial stages of aborted programmed cell death, thus leading to the fragmentation of chromatin. From the molecular point of view, this event ought to result in fatal consequences, i.e., the death of the majority of cells, although the remaining cell population that has survived would be forced to undergo the DNA reparation process. The cells that survive the aforementioned process (but do not perform proper reparation of previously cleaved DNA) are thought to be the origin of events related to chromothripsis. Since the changes undergoing in the molecular environment are random, this theory may be correct in terms of understanding the chromothripsis process. Throughout the process, exposed and cleaved chromatin regions might be a suitable and accessible target for the remaining chromosome fragments, regarding chaotic environmental conditions [19]. Therefore, the subsequent combining of DNA fragments might be obtained via the overall performance within the invalid repairment patterns, causing a cascade of random chromosomal rearrangements [20].

### 2.2. Prevalence of Chromothripsis

As the process of chromothripsis involves multiple, widespread genomic regions, its range is relatively broad within the genomic structure. Therefore, it is capable of inducing a significant number of various diseases [23]. This phenomenon is associated with congenital as well as cancer diseases; moreover, it has been proven that chromothripsis can be transmitted through the germline [24]. A previously proposed rate of chromothripsis oscillated between a 2 and 3% occurrence in various cancer (in blood cancers, central nervous system cancers, soft tissue tumours, and carcinomas), but even up to 25% of bone cancer (osteosarcoma and chordoma) [11]. In turn, Cortes-Ciriano and his group have presented the results frequency of chromothripsis in several types of cancer, stating even more than 50% events [25]. The group has evaluated the patterns characteristic for the chromothripsis in 2658 human cancers, using the WGS (whole genome sequencing) approach. The project has revealed that chromothripsis constitutes a major foundation for shaping the architecture of different cancer genomes. Results of the analysis have also undermined a former theory on the chromothripsis incidence rate in cancer diseases [25]. Earlier investigations based on the detection of copy number profiles were obtained from single nucleotide polymorphism (SNP) arrays that are more densely clustered than expected by chance, e.g., at least 10 adjacent CN alternations, i.e., in medulloblastoma [26]. WGS data provide an enhanced view of structural variations (SVs) and their type (deletion, insertion, inversion) in the genome with breakpoints identified at a single nucleotide resolution. Their analysis showed that the frequency of chromothripsis is over 40% for glioblastomas and lung adenocarcinomas. Furthermore, they affirmed that 100% of liposarcomas and 77% of osteosarcomas exhibit high-confidence chromothripsis. Additionally, ovarian adenocarcinomas, breast adenocarcinomas, melanomas, CNS-GBM, esophageal adenocarcinomas, and Lung-AdenoCA showed evidence of chromothripsis in >50% of the cases [25]. Chromothripsis can be considered as a potential prognostic marker indicating a higher severity of the disease and a poorer survival in neoplasms associated with chromothripsis [27]. These observations have been reported in diverse cancer types—mainly bone cancer, paediatric medulloblastoma (especially SHH-subtype), neuroblastoma, melanoma, colorectal cancer, and haematological malignancies [26,27,28,29,30,31,32,33]. A study conducted by Voronina et al. presents the analysis of chromothripsis in 28 tumour types covering all major adult cancers (634 tumours, 316 examined by whole-genome and 318 examined whole-exome sequences). They showed that chromothripsis affects a substantial proportion of human cancers, with a prevalence of 49% across all cases. Therefore, they suggested that the incidence of chromothripsis might have been underestimated using low-resolution methods. Notably, they identified pathogenic germline variants cancer predisposition genes including, among others, DNA repair genes from mismatch repair (MSH2, MSH6, MLH1) and double-strand break repair (ATM, NBN, BRCA1, BRCA2). Approximately 40% of the tumours with germline variants exhibited somatic loss of the wild-type allele [34]. The occurrence of cth is not distinctive to only malignant tumours. The prevalence of this phenomenon is also observed in benign cancer subtypes, such as uterine leiomyoma. Uterine fibroid cells usually harbour a relatively high number of genomic alterations (13–42%, as reported). Unlike malignant tumours, chromothripsis in uterine fibroid cells is characterised by fewer breakpoints (20 or more) and a larger number of affected chromosomes (up to four). They do not feature *TP53* mutations or histological signs of malignancy. Furthermore, unbalanced chromothripsis has been observed in both the cultured and non-cultured fibroid cells [35].

Chromothripsis detected in healthy individuals does not remain without any effects. de Pagter et al. analysed the genomes of three families in which chromothripsis rearrangements were transmitted from a mother to her child. The chromothripsis in the mothers resulted in completely balanced rearrangements involving 8–23 breakpoint junctions across three to five chromosomes. The chromothripsis in the mother’s resulted in a completely balanced rearrangement (two of them did not show any phenotypic abnormalities). Two children presented complex copy-number changes. This resulted in gene-dosage changes, which are probably responsible for the severe congenital phenotypes of these two children. In contrast, the third child, who has a severe congenital disease, harboured all three chromothripsis chromosomes from his healthy mother, but one of the chromosomes acquired de novo rearrangements leading to copy-number changes.

These results indicated that the human genome can tolerate extreme reshuffling of chromosomal architecture, including the breakage of multiple protein-coding genes, without visible phenotypic effects. By altering the function of gene-encoding proteins, it can be transmitted to the offspring, thus causing severe congenital abnormalities. Apart from the occurrence of congenital diseases, further implications of hereditary cth include a higher risk of miscarriages, or developmental delay [36]. Prevalence of chromothripsis in different disease entities presented in Figure 2.

*TP53*-deficient cells have successfully been used as an in vitro model of chromothripsis. The prevalence of germline variations of *TP53* have also been linked to paediatric SHH (Sonic Hedgehog) medulloblastoma and leukaemia subtypes. The analysis conducted by Rausch et al. has presented an unexpected germline in *TP53* mutants, suggesting that they were derived via rapid, massive shattering and chromosomal rearrangements [28]. The WGS approach, coupled with microarray analysis results, have subsequently been compared with deep sequencing-based DNA rearrangement data obtained from additional patients. All cases (10) of SHH-MB with altered *TP53* displayed rearrangements compatible with chromothripsis hallmarks. The evidence for chromothripsis could not been proven in the *TP53* (+) patients. Statistical analysis (two-tailed Fisher’s test) indicated that the mutation in the *TP53* gene is strongly associated with SHH-MB- derived chromothripsis (*p* = 1.6 × 10^−8^). One of the major conclusions drawn from the study was the fact that oncogenes associated with medulloblastoma are undergoing the frequent amplification process driven by chromothripsis. Oncogenes involved in Sonic Hedgehog signalling displayed significant enrichment which was limited to the regions affected by chromothripsis [28]. Neuroblastoma is a childhood cancer of the peripheral sympathetic nervous system. Frequently detected gene alterations are limited to *MYCN* amplification (20%) and *ALK* activations (7%). In his study, Molenaar et al. presented a WGS analysis of 87 neuroblastomas of all stages. The analysis of structural defects identified a local shredding of chromosomes, known as chromothripsis, in 18% of high-stage neuroblastoma. These tumours are associated with a poor outcome. Structural alterations recurrently affected *ODZ3*, *PTPRD* and *CSMD1*, which are involved in neuronal growth cone stabilisation. Furthermore, *ATRX*, *TIAM1* and the Rac/Rho pathway were mutated, further implicating defects in neuritogenesis in neuroblastoma. Most tumours with defects in these genes were aggressive high-stage neuroblastomas, but did not carry *MYCN* amplifications. The genomic landscape of neuroblastoma therefore reveals two novel molecular defects, chromothripsis and neuritogenesis gene alterations, which frequently occur in high-risk tumours [29].

## 3. Chromothripsis in Haematological Malignancies

The phenomenon of chromothripsis is significantly rare among the haematological malignancies and the complexity of chromothripsis events in haematological malignancies is less complex than in solid cancer. Cth has been observed in lymphomas, multiple myeloma, myelodysplastic syndrome, and leukaemias [25,32,33].

### 3.1. Multiple Myeloma (MM)

Analysis of SNP array data from 764 newly diagnosed multiple myeloma patients identified chromothripsis in 1.3% of the samples. Moreover, this catastrophic event confers a poor outcome. Magrangeas et al. have suggested on the basis of their results that high-risk MM patients use this novel way of cancer evolution [33]. The use of whole-genome sequencing has led to the identification of massive-genome rearrangement patterns in a number of myeloma patients [34,38]. Rustad et al. reported a 24% prevalence of chromothripsis, making multiple myeloma the haematological cancer with the highest documented prevalence of chromothripsis. MM patients with chromothripsis have presented poor clinical outcomes. It should be mentioned that all patients, apart from chromothripsis, had other multiple unfavourable clinical and genomic prognostic factors, including translocations involving *MAF*, *MAFB*, and *MMSET*, del17p13 and *TP53* mutations [39]. These observations have been confirmed by Maclachlan et al. They reported that chromothripsis was a detectable phenomenon in up to 30% of newly diagnosed myeloma patients. The detection of chromothripsis can redirect the route of therapy, since it is associated with relatively short overall survival (OS) in patients. Chromothripsis constitutes a new, independent prognostic factor for multiple myeloma, hence it is detectable years before the actual diagnosis. As Maclachlan et al. reported, copy number variation numbers are strongly predictive of chromothripsis in multiple myeloma patients [40].

The principals of cancer development have perpetually acknowledged the fact that oncogenic alterations are acquired throughout multiple changes occurring in extended periods of time. The minor changes that lied upon the pathophysiology of cancer were ought to be gathered gradually, and for the many years, the Darwinian theory of evolution was applicable in the explanation of the cancerogenesis process. Mutational and selectional-based changes were considered as the reason of disintegration within the critical genes, thus leading to the cancer onset [41]. This particular mechanism is specific to, i.a., multiple myeloma, where the onset is preceded by the sustained proliferation of benign plasma cells. Despite the validity of this theory, years of research have proven the fact that it may not be an exclusive reason for the loss of homeostasis associated with the malignancy progress. In contrast to solid cancers, in MM, chromothripsis represents an early driver event detectable years before the diagnosis, which remains relatively stable over time. Moreover, chromothripsis is emerging as one of the strongest features able to predict both the progression of the myeloma precursor condition to MM and shorter OS with MM, independent of other known prognostic variables [38,40].

### 3.2. Chronic Lymphocytic Leukaemia (CLL)

Chronic lymphocytic leukaemia (CLL), the most prevalent adult leukaemia, is characterised by a highly variable clinical course. Stephens and his colleagues have discovered multiple somatically acquired genetic variants (chromothripsis) in patients with chronic lymphocytic leukaemia. The sequencing method implemented in this particular investigation enabled them to screen the genome-wide rearrangements. As a result of the successful screening process, the CLL patient with 42 somatic genomic rearrangements was selected. The aforementioned rearrangements involved the long arm of chromosome 4 (4q), being a relatively large part of the genome (approximately 140 Mb), making it more than the whole genome capacity of chromosome 10. Apart from a separate 13q deletion in this patient, all rearrangements are confined to the long arm of chromosome 4 and focal points on chromosomes 1, 12, and 15. The aforementioned genome alterations were detected in the sample from the patient prior to treatment implementation. The CLL patient was administered with alemtuzumab—treatment was followed with her unfortunate, quick relapse. In order to assess whether the alterations were present in the cells that have undergone anti-CLL regimen, Stephens and his colleagues have sequenced the sample collected 31 months after initial, pre-treatment sampling; analysis confirmed the presence of all preceding alterations. Conversely, the absence of emerging, additional genomic changes was an anchor point to the acknowledgement that the complex changes and the genome remodelling processes have arisen before the initial screening of the patient [25]. Bassaganyas et al. have described a 59-year-old *IGHV* unmutated CLL patient and over 11 years of clinical evolution. Conventional cytogenetic and fluorescence in situ hybridisation analysis showed only a derivative chromosome formed by 4q (duplicated) and 18q (with 18p deletion) but an SNP array and targeted re-sequencing have revealed 49 SVs detected by WGS. Most SVs showed chromosome geographical localisation, with 80% of breakpoints (involving 35 of 49 SVs) confined to focal points on chromosomes 1, 6, 10 and 12 and demonstrated the existence of complex genomic rearrangements consistent with the pattern of chromothripsis. They have analysed three samples from the same patient obtained at different time points, wanting to indicate the potential role of SNVs in the CLL initiation. Their observations revealed that chromothripsis subclones did not survive chemotherapy and that it has not reappeared in a period of 10 years. Therefore, they concluded that chromothripsis in CLL have no implications for patient prognosis [42]. This differs from previous data strongly associating chromothripsis with chemotherapy surviving and/or poor clinical outcome [15,27,38,40]. A study conducted by Salaverria et al. revealed a chromothripsis-like pattern in eight cases on 180 samples CLL patients (DNA patients were hybridised to the qChip^®^Hemo array). Three showed concomitant shattered 5p with a gain of TERT along with isochromosome 17q. The presence of 11q loss was associated with a shorter time to first treatment, whereas 17p loss increased the genomic complexity, and chromothripsis was associated with a shorter overall survival [43]. Another study has reported data coming from the retrospective analysis of 2293 arrays from 13 diagnostic laboratories. Complex patterns (n = 32), indicative of chromothripsis, were also detected. Cases with cth were all del(11q) or *TP53*abn-positive, and chromosomes 2, 3, 6, 8 and 17 were mainly targeted of cth. Chromothripsis was associated with an adverse outcome (median OS: 3.7 years) and a high total number of CNA (median 6). The survival of cases with chromothripsis was also worse than that of cases with *TP53*abn or del(11q) without chromothripsis [44]. Patterns suggestive of chromothripsis or chromothripsis-like were identified in 30 patients on 340 giving prevalence of cth of 8.3% in CLL in an investigation conducted by Ramos-Campoy et al. A negative impact was observed for chromothripsis patients (cth was associated with complex karyotype and *TP53* alterations) who showed a short time to first treatment (TTFT) [45]. Concluding highly complex karyotypes associated with cth can drive the development of CLL. CLL patients harbouring chromothripsis have demonstrated an inferior overall outcome (first-line treatment response, progression-free survival and overall survival rate). The presence of chromothripsis has been associated with phenotypically severe genomic aberrations, such as deletions of 11q or 17p. Some studies suggest CLL may be driven by chromothripsis— either cth may be an initiating factor, or it can partially contribute to early events contributing to the development of CLL [44,45].

### 3.3. Acute Myeloid Leukaemia (AML)

In the course of AML, with a complex aberrant karyotype, a loss of one *TP53* allele is frequently observed. Haferlach C et al. analysed the incidence of *TP53* mutations and deletions in 107 AML with a complex aberrant karyotype. They confirmed a high incidence of *TP53* mutations in AML with a complex aberrant karyotype (29/42, 69%) and demonstrated that *TP53* mutations are very rare in AML without a complex aberrant karyotype (4/193, 2.1%). It has been proven that the arising complexity of karyotypes in AML may arise from acquired mutations of *TP53* [46]. Another integrative analysis performed by Rucker et al. proved the prominent association of *TP53* impairments with the chromothripsis process. One hundred and twelve complex karyotype-AML (CK-AML) patients have undergone the genomic profiling process, followed by *TP53* mutation screening and GEP (global gene expression) performed in a certain number of cases. The chromosomal distribution of chromothripsis was distributed among all of the autosomes, with the highest prevalence in chromosome 3, 7 (10%) and chromosome 12 (9%). SNP profiling detected chromothripsis-like rearrangements characterised by numerous (at least ten) switches between 2/3 CNS, limited to individual chromosomes in 35% of the cases. The entire group harbouring *TP53* mutations did not demonstrate excessive rearrangements consistent with chromothripsis patterns. To gain insight into *TP53*-derived genomic alterations, the group performed explorative subset analysis for genes associated with *TP53*. Chromothripsis-positive, and subsequently, -negative groups were compared in order to assess the gene signature for *TP53* altered and chromothriptic subjects. Among the group of 337, *PSMB10* and *BCL9* (characterised by fold change) genes were proven to be the most deregulated. *PSMB10* belongs to the proteasome-regulating gene family and its expression is elevated in the majority of cancer types [47]. The B-cell CLL/lymphoma 9 protein, encoded by *BCL9*, is involved in the Wnt pathway which is essential in the tumour progression process. CK-AML altered *TP53* demonstrated also a major dysregulation of genes engaged in genomic instability induction (i.e., *ERCC1, FANCA* or *MLH1*) associated with Fanconi Anaemia, thereby explaining the link between telomere dysfunction and the onset of chromothripsis [47]. Fontana et al. reported a 6.6% chromothripsis in AML in their study [32]. Patients with AML harbouring cht were characterised by a higher age, lower WBC count, mutually exclusive with *FLT3* and *NPM1* mutations and a loss and/or mutations of the *TP53* gene. Subsequent genomic characteristics involved rearrangements on chromosomes 12, 17, 5, 6, 3, 4, 7, 10, 11, 15 and 20. SNP microarray analysis has proved that cht (+) patients displayed a relatively high grade of aberrations within the genome. The presence of chromothripsis-like rearrangements has been associated with the formation of chromosomal derivatives: marker, ring or derivative chromosomes resulting from unbalanced translocations. Moreover, the study defined a group of AML patients with a poor overall survival rate, among which chromothriptic patients demonstrated a specifically low survival rate. They concluded that chromothripsis was associated with losses of 5q31.1–5q33.1 in most patients, and with a complex genomic background in which *FANCA*, *TP53*, and genes regulating cell cycles seem to be fundamental and demand further preclinical studies [32]. The analysis conducted by Bochtler et al. included all patients enrolled into the 2 large consecutive, prospective, randomised, multicentre AML96 and AML2003 trials of the Study Alliance Leukaemia by array comparative genomic hybridisation. Almost one-third of marker chromosomes (18/49) were derived from chromothripsis, whereas this phenomenon was undetectable in a control group of marker chromosome-negative complex aberrant karyotypes (1/34). They suggested that the chromothripsis-positive cases were characterised by a particularly high degree of karyotype complexity, *TP53* mutations, and dismal prognosis, and they concluded that marker chromosomes are indicative of chromothripsis and associated with poor prognosis and not merely by an association with other adverse cytogenetic features [48]. Mackinnon et al. introduced the term anachromosome to describe an abnormal chromosome produced by chromothripsis. The anachromosomes conformed to the normal constraints of the chromosome structure by including segments that provide two telomeres and a centromere. They described two interesting cases of acute myeloid leukaemia matching the description of chromothripsis that illustrate different aspects of this phenomenon from a cytogenetic perspective. An 81-year-old male with at least 11 deletions of chromosome 1, including a deletion of the 1q telomeric section and a single, terminal deletion of 18q, and chromosome 1 and 4 underwent intrachromosomal rearrangements, with the exception of the joining of chromosome 18 to the repaired chromosome 1 regions. The second case is a 28-year-old female with the monosomies chromosomes 2, 5, 14, 17 and in G-banding. However, comprehensive analysis, M-FISH, M-BAND, FISH and microarray, revealed at least seven separate deletions on chromosome 1 and eight separate deletions on chromosome 5, suggesting that both chromosomes had undergone chromothripsis. In case 1 and case 2, six broken chromosomes were involved in a complex web of rearrangements. Despite the involvement of six chromosomes in the rearrangements of case 2, this does not appear to fit the description of chromoplexy, which appears to be a complex unbalanced rearrangement event involving several chromosomes as most of the breakpoints and rearrangements involved just two of these chromosomes [49]. It appears that a critical combination of deleted and retained regions is sometimes more oncogenic than simple monosomy, and chromothripsis could facilitate such a rearrangement [50]. The deletion of 5q, 17p, and 20q stimulates AML and myelodysplastic syndromes (MDS), but the frequency of complete monosomies of these chromosomes are rare [51,52]. Abaigar et al. conducted the study exploring novel genetic abnormalities occurring in MDS by aCGH and NGS. Three hundred and one patients diagnosed with MDS (n = 240) or MDS/MPN (n = 61) were studied at the time of diagnosis. Three high-risk MDS cases (1.2%) displayed chromothripsis involving exclusively chromosome 13 and affecting some cancer genes: *FLT3*, *BRCA2* and *RB1*. All three cases carried *TP53* mutations, as revealed by NGS. In addition to well-known copy number defects, the presence of chromothripsis involving chromosome 13 was a novel recurrent change in high-risk MDS patients [53].

### 3.4. Acute Lymphoblastic Leukaemia (ALL)

In the study of Forero-Castro et al., genome-wide DNA copy number analysis allows the identification of genetic markers that predict the clinical outcome, suggesting that the detection of these genetic lesions will be useful in the management of patients newly diagnosed with ALL. Array-CGH was carried out in 265 newly diagnosed ALLs (142 children and 123 adults). Three of the patients showed chromothripsis (cth6, cth14q and cth15q). CNAs were associated with age, phenotype, genetic subtype and overall survival (OS). In the whole cohort of children, the losses on 14q32.33 and 15q13.2 were related to shorter OS [54]. Ratnaparkhe et al. have used three techniques: whole-genome sequencing, fluorescence in situ hybridisation and RNA sequencing to characterise the genomic landscape of ALL patients with Ataxia Telangiectasia. They identified a high frequency of chromothriptic events in these cancers, specifically on acrocentric chromosomes, as compared with cancers from individuals with other types of DNA repair syndromes (27 cases total, 10 with Ataxia Telangiectasia) [55]. They suggested that the genomic landscape of Ataxia Telangiectasia ALL is clearly distinct from that of sporadic ALL. Ratnaparkhe and colleagues also hypothesised that acquired germline *ATM* mutations are linked with the prevalence of Cht in ALL patients—any of the non-Cht sporadic cases did not exhibit mutated *ATM*, which is known to be an essential guardian of genome stability and it prevents DNA damage, i.a., deleterious lesions. It is proven that *ATM* is engaged in telomere-mediated cellular processes and its deficiency can induce the formation of dicentric chromosomes prone to chromothriptic events, as proven in iAMP21 cases. To examine the association between the *ATM* state and telomere dysfunction, they measured telomere length in the germline and tumour cells and executed the comparison between two groups: (I) A-T patients and NBS (Nijmegen breakage syndrome) and (II) ALL with A-T and sporadic ALL. Telomere length was significantly shorter in A-T cases, with no significant difference between ALL (T-A) cases with or without Cht. For further investigations, they analysed the number of breakpoints per single chromosome and the CN-state of the *ATM* locus, examining public datasets of various types of cancer (lung, stomach, bladder, cervical, colon). Obtained results have established that the loss of *ATM* is linked to an increased number of breakpoints per chromosome, proving the relevance of *ATM* in the progression of the cell cycle [55].

Turner et al. have studied amplified driver oncogenes which, in cancer, can be amplified in chromosomes or in circular extrachromosomal DNA (ecDNA), although the frequency and functional importance of ecDNA are not understood [56]. The over-expression of oncogenes or the increasing expression of a gene whose action diminishes the efficacy of an anti-cancer drug can help develop cancer therapy resistance. Shoshani et al. have used WGS of clonal isolates developing chemotherapeutic resistance to identify chromothripsis as a cause of the arising of extrachromosomal DNA (ecDNA) amplification into circular double minutes (DMs). They concluded that chromothripsis is a primary mechanism accelerating genomic DNA amplification and which enables the rapid acquisition of tolerance to altered growth conditions [57]. In terms of mechanisms driving genome rearrangement, Rosswog et al. have presented a novel type of amplification—seismic amplification. The hallmarks of this phenomenon are: the presence of multiple rearrangements and the discontinuity of copy number levels. Data obtained from paediatric neuroblastoma patients have enabled the discovery of this mechanism; however, whole genome analysis performed in 2756 individuals has proven that seismic amplification is prevalent in 9.9% of cases across 38 types of cancer. Seismic amplification is characterised by high-level alterations and an elevated gene expression. In order to gain an insight into the developmental patterns of these alterations, WGS data of 79 neuroblastoma patients were examined. The amplification patterns present in the samples differed from chromothripsis, since they harboured a limited number of oscillating CN states. Observations obtained via the FISH approach revealed cases of seismic amplifications occurring in an extrachromosomal as well as intrachromosomal manner. Further optical mapping has demonstrated duplications of previously rearranged genomic fragments. This finding suggests the involvement of genomic recombination in the seismic amplification process, indirectly linking it to cht-based rearrangements. Despite the differences between seismic amplification and cht, it is a common event leading to focal rearrangements in cancer and it may contribute to seismic amplification. Phenomena supporting this concept are, correspondingly: the presence of the clustering of rearrangement breakpoints, the distribution patterns of various rearrangement types or the alternating character of losses/ amplifications present in seismic amplification [58].

One of important new provisional entities which has been recognised in B cell acute lymphoblastic leukaemia/lymphoma (B-ALL) is the intrachromosomal amplification of chromosome 21 (iAMP21). iAMP21 was defined as three or more extra copies of RUNX1 on a single abnormal chromosome 21 (a total of five or more RUNX1 signals per cell) with a complex structure comprising multiple regions of gain, amplification, inversion and deletion. Intrachromosomal amplification within the 21 chromosome is strongly linked to a dismal therapy outcome and survival rate in B-ALL when treated on standard therapy, as compared with other patients treated on the same protocols. However, when iAMP21 patients treated as high risk showed an improved outcome regardless of the backbone chemotherapy regimen given, indicating iAMP21 patients will be treated as cytogenetic high risk, they received intensive chemotherapy. iAMP21 is characteristic for paediatric patients and the sole mechanism (iAMP) emerges from the BFB cycle and genomic rearrangements characteristic for chromothripsis. Accounting for 2% of paediatric B-ALL cases, this phenomenon is relatively rare among this particular condition. iAMP21 patients generally had low white cell counts (WCCs) with a median age of 9–11 years [59,60]. Gu et al. have previously reported a rare case of paediatric B-ALL accompanied by chromothripsis. An 18-year-old Caucasian man was diagnosed with *ETV6-RUNX1* (+) B-ALL. FISH analysis and G-band karyotyping have revealed complex chromosomal abnormalities with iAMP21, ETV6-RUNX1 fusion and a *TP53* mutation. Despite the implementation of intensive chemotherapy combined with allogenic SCT, the patient died 26 months after initial diagnosis. The researchers also identified six other cases of B-ALL with a co-occurrence of iAMP21 and/or *ETV6/RUNX1* fusion. One of the patients demonstrated the *ETV6/RUNX1* karyotype at the initial diagnosis and acquired iAMP21 (at the relapse state), which indicates that massive chromosome rearrangements may underlie the development of intrachromosomal amplification within chromosome 21 [61]. Li et al. used genomic, cytogenetic and transcriptional analysis, with novel bioinformatic approaches, to reconstruct the evolution of iAMP21 ALL. They found that children with the rare constitutional Robertsonian translocation between chromosomes 15 and 21, (rob(15;21)(q10;q10)c), have ~2700-fold increased risk of developing iAMP21 ALL compared to the general population. Probably, the amplification is inducted by a chromothripsis event involving both sister chromatids of the Robertsonian chromosome. This study showed that cth can causes a loss of multiple, non-contiguous chromosomal regions, and that whole chromosome duplication gives expansive, low-amplitude amplification [62]. A report from Rode et al. has involved the hallmarks and prevalence of cht among different tumour entities. According to results obtained via a whole genome sequencing approach, the prevalence of cht in ALL accompanied by iAMP21 is relatively high (prevalence (%)^2^ reaching 88.9) [37].

### 3.5. Chromothripsis-like Events in Congenital Disorders

Chromothripsis-like events have been associated with congenital disorders and mental retardation cases, proving chromothripsis is not distinctive to merely cancer genome and carcinogenesis. Congenital chromothripsis has been found in the paternal germlines, however, this fact does not undermine the possibility of primary occurrence of chromothripsis during early embryogenesis [3,23,24]. Chromothripsis-like rearrangements occurring in the germline have subsequently been reported. These cases are characterised by a limited range of alterations—the majority of reported subjects harboured rearrangements involving one chromosome. Since the rearrangements discovered in the germline are characterised by fewer numbers of breakpoints, a lower complexity level is considered one of the distinctive hallmarks for these rearrangements (compared to the alteration patterns in cancer-derived chromothripsis). Germline chromoanagenesis-like events could be induced by environmental factors such as increased parental age, often associated with the decreased ability of DNA repair machinery [3]. Phenotypically, congenital chromothripsis manifests itself in mental retardation and dysmorphia which may (but does not have to) be accompanied by additional clinical features. Complex chromosomal alterations occurring in the germline are not necessarily pathogenic, however, they can affect multiple genes regarding their copy-number state and exert deletions which may impact overall cell growth and differentiation [63]. Ovarian dysfunction is associated with the chromothripsis process. This particular disease is acquired via rearrangements within the X chromosome. However, in the majority of women harbouring the rearrangements in the X chromosome, the affected chromosome undergoes inactivation. Complex chromosomal rearrangements in the X chromosome have also been associated with a case of developmental delay involving short stature (the girl carried an unmasked recessive mutation in *CSF2RA* gene) [6].

## 4. Diagnostics of Chromothripsis

A diagnosis of chromothripsis can be conducted in the prenatal stage at the earliest. The first written mention of the unambiguous detection of chromothripsis was published by Nazaryan et al. in 2014. The paper emphasised the necessity of implementing several complementary methods (such as FISH, karyotyping and Sanger sequencing) in order to obtain the whole picture of complex rearrangements [64,65]. The topic of complex cth detection reappeared in 2016, when Macera et al. published a report of chromothripsis diagnostics via multiple molecular biology approaches: karyotyping, fluorescence in situ hybridisation (FISH), microarray, whole genome sequencing approach. These methods can help determine the organisation of abnormal genomes after chromothripsis and other types of complex genome rearrangement. At this time, the complexity of chromothripsis phenomenon required using the cytogenetic and molecular testing simultaneously [66]. The method of karyotyping by G-banding is used to identify numerical and structural chromosomal abnormalities, but the resolution of the light microscope and high labor intensity are the limitations of this method. Many of these details can be determined by the strategic use of metaphase FISH. FISH is a single-cell technique, so it can identify low-frequency chromosome abnormalities, and it can determine which chromosome abnormalities occur in the same or different clonal populations. These are important considerations in cancer. Metaphase chromosomes are intact, so information about abnormalities of the chromosome homologues is preserved. The spectral karyotyping (SKY) and multicolour FISH analysis (M-FISH) are a particularly informative kind of FISH method to analyse cth. DNA microarrays can identify the changes in the copy number, but they do not give information on the organisation of the abnormal chromosomes, balanced rearrangements, or abnormalities of the centromeres and other regions comprised of highly repetitive DNA. NGS is one of the most effective methods for detecting structural variations in the genome. The method is based on the amplification of a multitude of short sections of genes, in their totality, covering the entire genome, followed by their sequencing. This method makes it possible to identify both numerical and structural alterations in the genome. However, NGS has its limitations, namely, the method cannot identify copy-neutral options and breakpoints for structural alterations [17,67]. The RNA-Seq approach enables the detection and characterisation of a *fusion* or *chimeric* transcript associated to a complex genome rearrangement [67]. To better understand the molecular mechanisms of cth, an easy to use tool (CTLPScanner) was presented for automatically detecting a chromothripsis-like pattern (CTLP) in genomic array data. They integrated over 50,000 pre-processed genomic arrays from The Cancer Genome Atlas and Gene Expression Omnibus into the CTLPScanner to assist in performing meta-data analysis. The CTLPScanner was designed for users to screen chromosome pulverisation regions and obtain annotations. It assesses the most striking features of chromothripsis, i.e., clustering of chromosomal breakpoints and oscillating copy number changes in specific regions. The CTLPScanner will help reveal more general features of this new paradigm in tumourigenesis, and thus eventually improving cancer classification [68].

## 5. Conclusions

Somatic abnormalities within the human genome are characterised by continuous changes. They may be subtle and develop in a complex manner, or can be limited to a single, catastrophic event affecting a considerable part of the genome. Chromothripsis, along with the remaining chromoanagenesis-derived phenomena, is representing a clinically relevant process of massive genome restructuring. The impact of chromothripsis, as well as massive SVs on the medical practice, is major—further research of the cth phenomenon may lead to the identification of predictive biomarkers of treatment response. Therefore, in our opinion it will be necessary to use an integrated approach that will include several methods of genetic analysis at once as the best option to diagnose chromothripsis. A very interesting issue is the occurrence of chromothriptic events in children’s neoplasm (iAMP21 as a chromothripsis-like event) and their role in the prediction of the outcome. We are aware that additional studies are definitely needed with the attraction of a large amount of data to use this type of aberration as a prognostic factor in anticancer therapy. Maybe, a potential therapy of this phenomenon will implicate a combined treatment involving targeted therapy along with the use of DNA repair inhibitors (in order to prevent subsequent rearrangements driven by chromothripsis).

## Figures and Tables

**Figure 1 ijms-23-03318-f001:**
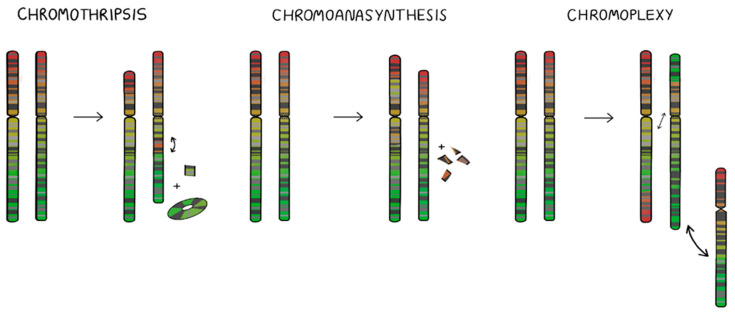
Schematic overview of three chromosomal types of catastrophic events, known as chromoanagenesis.

**Figure 2 ijms-23-03318-f002:**
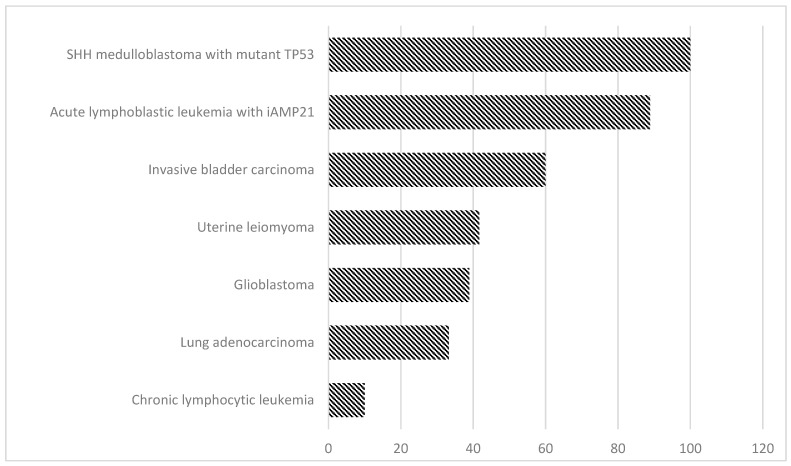
Prevalence of chromothripsis analysed by WGS in different disease entities, according to Rode et al. [37].

## Data Availability

Not applicable.

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
