# Peer review of "Insight into the Molecular Basis Underlying Chromothripsis"

_ijms, 2022, doi:10.3390/ijms23063318_

Round 1

Reviewer 1 Report

The authors presented in great detail the cases of chromothripsis in various types of  haematological malignancies. However, based on the name "Insight into the molecular basis lying under chromothripsis", I would like to know the opinion of the authors themselves on the molecular basis of chromothripsis, made on the basis of a review of a number of works. There is a lack of description and discussion of the presence of chromotripsis not only in malignant, but also in benign tumors, as well as in the karyotype of healthy individuals. (Koltsova A S., Pendina A A., Efimova OA., Chiryaeva O G., Kuznetzova T V., Baranov V S. On the Complexity of Mechanisms and Consequences of Chromothripsis: An Update.  Frontiers in Genetics; 2019; 10. DOI=10.3389/fgene.2019.00393). Chromotripsis in these cases cannot be unequivocally associated with an unfavorable prognosis. Generally, the review is quite detailed, but needs to be improved.

Author Response

Response to Reviewer 1 Comments:

Dear Sir or Madam, thank you very much for the review of our manuscript entitled:  Insight into the molecular basis underlying chromothripsis.

In response to your comment, we would like to thank you for appreciating our manuscript.

The authors presented in great detail the cases of chromothripsis in various types of haematological malignancies. However, based on the name "Insight into the molecular basis lying under chromothripsis",

Comment 1

I would like to know the opinion of the authors themselves on the molecular basis of chromothripsis, made on the basis of a review of a number of works.

Revision and my comments

The information was added in the conclusion.

Comment 2

There is a lack of description and discussion of the presence of chromotripsis not only in malignant, but also in benign tumors, as well as in the karyotype of healthy individuals.

(Koltsova A S., Pendina A A., Efimova OA., Chiryaeva O G., Kuznetzova T V., Baranov V S. On the Complexity of Mechanisms and Consequences of Chromothripsis: An Update.  Frontiers in Genetics; 2019; 10. DOI=10.3389/fgene.2019.00393).

Chromotripsis in these cases cannot be unequivocally associated with an unfavorable prognosis. Generally, the review is quite detailed, but needs to be improved.

Revision and my comments

The new information was added in the manuscript (paragraph prevalence chromothripsis) according to the Reviewer’s comment.

Best regards

Monika Lejman

Reviewer 2 Report

Review of a manuscript entitled: "Insight into the molecular basis lying under chromothripsis”, by Monika Lejman and Katarzyna OstapiÅ„ska (IJMS-1621531).

This manuscript reviews chromoanagenesis, chromoanasynthesis and chromothripsis in the germline and in somatic cells involved in malignant disease. Molecular mechanisms of chromoanagenesis events, chromothripsis in malignant disease and in development are reviewed. While the sections on chromothripsis in malignant disease are exhaustive, the other parts of this review need a few additions and some retructuring as detailed below.

In line 41 the authors should add to reference 2 also: Poot M. Genes, proteins and biological pathways preventing chromothripsis. Methods in Molecular Biology 1769: 231-251 (2018), which is a comprehensive review of the possible mechanisms underlying chromoanagenesis, chromoanasynthesis and chromothripsis up to 2018.

In the paragraph on Chromoanasynthesis in the introduction (lines 42 through 55) the authors should also mention the review by Carvalho and Lupski. Mechanisms underlying structural variant formation in genomic disorders. Nat Rev Genet 17(4): 224–238 (2016).

The reference 7 in line 60 is a comment on the research article listed as reference 8. It is better replaced by reference 57 the first description of chromoplexy at all (Kloosterman WP, Guryev V, van Roosmalen M, Duran KJ, de Bruijn E, Bakker SC, Letteboer T, van Nesselrooij B, Hochstenbach R, Poot M, Cuppen E: Chromothripsis as a mechanism driving complex de novo structural rearrangements in the germline. Hum Mol Genet 20: 1916–1924 (2011) as has been pointed out by Zhang, Leibowitz and Pellman: Chromothripsis and beyond: rapid genome evolution from complex chromosomal rearrangements. Genes Dev 27:2513–2530 (2013). The publication by Kloosterman et al. on germline chromoplexy preceded the one by Baca et al.

The paragraph from line 120 on and the one from 160 through 176: abortive apoptosis involves the formation of one or several micronuclei. It is therefore conceivable that micronucleus formation, in particular in tumors, represents a mild form of abortive apoptosis. In the experimental system in which chromothripsis arose after micronucleus formation (Zhang et al., reference 19) TP53 had to ablated in order to allow the cells to proceed past the cell cycle checkpoint and to undergo chromothripsis (see Zhang, Leibowitz and Pellman: Chromothripsis and beyond: rapid genome evolution from complex chromosomal rearrangements. Genes Dev 27:2513–2530 (2013) and Poot M. Genes, proteins and biological pathways preventing chromothripsis. Methods in Molecular Biology 1769: 231-251 (2018). Thus, ablation of cell cycle checkpoints, such as TP53, is key to chromothripsis by either micronucleus formation or by the breakage-fusion-bridge mechanism. The authors should try to rearrange these paragraphs such that loss of cell cycle checkpoint control, abortive apoptosis and micronucleus formation appear in a logical order.

In line 138 do the authors mean centromeres or telomeres?

Line 182: the reference to the review by Kloosterman and Cuppen should be replaced by the original demonstration of widespread germline chromothripsis: Kloosterman WP, Tavakoli-Yaraki M, van Roosmalen MJ, van Binsbergen E, Renkens I et al: Constitutional chromothripsis rearrangements involve clustered double-stranded DNA breaks and nonhomologous repair mechanisms. Cell Rep 1: 648–655 (2012).

In the paragraph from line 268 on the authors seem to hint at the `double hit` hypothesis of the late Alfred Knudsen. If so, the authors should insert a reference, maybe: Loeb LA: A mutator phenotype in cancer. Cancer Res. 61: 3230–3239 (2001)

Line 517: reference 57 should be replaced by Kloosterman WP, Tavakoli-Yaraki M, van Roosmalen MJ, van Binsbergen E, Renkens I et al: Constitutional chromothripsis rearrangements involve clustered double-stranded DNA breaks and nonhomologous repair mechanisms. Cell Rep 1: 648–655 (2012), Poot M. Genes, proteins and biological pathways preventing chromothripsis. Methods in Molecular Biology 1769: 231-251 (2018), and reference 20, since reference 57 deals with a single patient (male) with a case of chromothripsis subsequent to chromoplexy, which does not allow any generalizations as opposed to Klooesterman et al., 2012, Narazyan-Petersen and Tommerup, 2016, and Poot, 2018.

In line 525: Poot M. Genes, proteins and biological pathways preventing chromothripsis. Methods in Molecular Biology 1769: 231-251 (2018) should be added to reference 58, since Poot, 2018 extensively discusses DNA repair defetcs in relation to chromothripsis.

Line 529: here the authors should also include: Pellestor F, Anahory T, Lefort G, Puechberty J, Liehr T, et al: Complex chromosomal rearrangements:origin and meiotic behavior.Hum Reprod Update 17: 476–494 (2011), since this reference for the first time shows that the majority of cases of complex chromotosme rearrangements are without phenotypic consequences.

The statement in line 530 “Ovarian dysfunction is associated with chromothripsis process” needs a reference.as well as the one in line 531.

Lines 537 through 541 should also include Nazaryan L, Stefanou EG, Hansen C, Kosyakova

N, Bak M, et al: The strength of combined cytogenetic and mate-pair sequencing techniques

illustrated by a germline chromothripsis rearrangement involving FOXP2 . Eur J Hum Genet 22: 338–343 (2014), who were the first to empasize the need for several complementary methods to unambiguously detect chromothripsis.

The statements in lines 545 through 554 need some elaboration. How do the authors infer chromothripsis from micronucleus formation (or abortive apoptosis) and gene amplification?

In this section the authors should also discuss: Marcozzi A, Pellestor F, Kloosterman WP. The origin and characteristics of chromothripsis, Methods in Molecular Biology 1769: ??-?? (2018) and Yang J, Liu J, Ouyang L, Chen Y, Liu B, Cai H. CTLPScanner: a web server for chromothripsis-like pattern detection. Nucleic Acids Res 44:W252-W258 (2016).

Minor point

As a rule authors who are not born and educated in an English speaking country should seek advice of a native speaker of English. The currect manuscript is a case in point. The authors may unknowingly have introduced statements that are difficult to grasp. Just one example: in the title the authors state “lying under”, which in English has only the literate meaning of being physically underneath. The authors may have meant the figurative meaning of “being a cause, giving rise to ...” etc. Also the use of articles (the, a) is unusal. Therefore this manuscript needs thorough desk editing.

Author Response

Response to Reviewer 1 Comments:

Dear Sir or Madam, thank you very much for the review our manuscript entitled: 

Insight into the molecular basis underlying chromothripsis.

In response to your comment, we would like to thank you for appreciating our manuscript.

Review of a manuscript entitled: "Insight into the molecular basis lying under chromothripsis”, by Monika Lejman and Katarzyna OstapiÅ„ska (IJMS-1621531).

This manuscript reviews chromoanagenesis, chromoanasynthesis and chromothripsis in the germline and in somatic cells involved in malignant disease. Molecular mechanisms of chromoanagenesis events, chromothripsis in malignant disease and in development are reviewed. While the sections on chromothripsis in malignant disease are exhaustive, the other parts of this review need a few additions and some retructuring as detailed below. 

 Comment 1

In line 41 the authors should add to reference 2 also: Poot M. Genes, proteins and biological pathways preventing chromothripsis. Methods in Molecular Biology 1769: 231-251 (2018), which is a comprehensive review of the possible mechanisms underlying chromoanagenesis, chromoanasynthesis and chromothripsis up to 2018. 

Revision and my comments

The reference was added as 3.

 Comment 2

In the paragraph on Chromoanasynthesis in the introduction (lines 42 through 55) the authors should also mention the review by Carvalho and Lupski. Mechanisms underlying structural variant formation in genomic disorders. Nat Rev Genet 17(4): 224–238 (2016).

Revision and my comments

The reference was added as 6.

 Comment 3

The reference 7 in line 60 is a comment on the research article listed as reference 8. It is better replaced by reference 57 the first description of chromoplexy at all (Kloosterman WP, Guryev V, van Roosmalen M, Duran KJ, de Bruijn E, Bakker SC, Letteboer T, van Nesselrooij B, Hochstenbach R, Poot M, Cuppen E: Chromothripsis as a mechanism driving complex de novo structural rearrangements in the germline. Hum Mol Genet 20: 1916–1924 (2011) as has been pointed out by Zhang, Leibowitz and Pellman: Chromothripsis and beyond: rapid genome evolution from complex chromosomal rearrangements. Genes Dev 27:2513–2530 (2013). The publication by Kloosterman et al. on germline chromoplexy preceded the one by Baca et al.

Revision and my comments

The references were changed as 9.

 Comment 4

The paragraph from line 120 on and the one from 160 through 176: abortive apoptosis involves the formation of one or several micronuclei. It is therefore conceivable that micronucleus formation, in particular in tumors, represents a mild form of abortive apoptosis. In the experimental system in which chromothripsis arose after micronucleus formation (Zhang et al., reference 19) TP53 had to ablated in order to allow the cells to proceed past the cell cycle checkpoint and to undergo chromothripsis (see Zhang, Leibowitz and Pellman: Chromothripsis and beyond: rapid genome evolution from complex chromosomal rearrangements. Genes Dev 27:2513–2530 (2013) and Poot M. Genes, proteins and biological pathways preventing chromothripsis. Methods in Molecular Biology 1769: 231-251 (2018). Thus, ablation of cell cycle checkpoints, such as TP53, is key to chromothripsis by either micronucleus formation or by the breakage-fusion-bridge mechanism. The authors should try to rearrange these paragraphs such that loss of cell cycle checkpoint control, abortive apoptosis and micronucleus formation appear in a logical order.

Revision and my comments

The paragraph was modified according to the Reviewer’s comments.

Comment 5

In line 138 do the authors mean centromeres or telomeres?

 Revision and my comments

The authors mean telomeres. We corrected.

Comment 6

Line 182: the reference to the review by Kloosterman and Cuppen should be replaced by the original demonstration of widespread germline chromothripsis: Kloosterman WP, Tavakoli-Yaraki M, van Roosmalen MJ, van Binsbergen E, Renkens I et al: Constitutional chromothripsis rearrangements involve clustered double-stranded DNA breaks and nonhomologous repair mechanisms. Cell Rep 1: 648–655 (2012).

Revision and my comments

The reference was added as 24.

Comment 7

In the paragraph from line 268 on the authors seem to hint at the `double hit` hypothesis of the late Alfred Knudsen. If so, the authors should insert a reference, maybe: Loeb LA: A mutator phenotype in cancer. Cancer Res. 61: 3230–3239 (2001)

Revision and my comments

The reference was added as 41.

 Comment 8

Line 517: reference 57 should be replaced by Kloosterman WP, Tavakoli-Yaraki M, van Roosmalen MJ, van Binsbergen E, Renkens I et al: Constitutional chromothripsis rearrangements involve clustered double-stranded DNA breaks and nonhomologous repair mechanisms. Cell Rep 1: 648–655 (2012), Poot M. Genes, proteins and biological pathways preventing chromothripsis. Methods in Molecular Biology 1769: 231-251 (2018), and reference 20, since reference 57 deals with a single patient (male) with a case of chromothripsis subsequent to chromoplexy, which does not allow any generalizations as opposed to Klooesterman et al., 2012, Narazyan-Petersen and Tommerup, 2016, and Poot, 2018.

Revision and my comments

The reference was added as 3, 24, 63.

 Comment 9

In line 525: Poot M. Genes, proteins and biological pathways preventing chromothripsis. Methods in Molecular Biology 1769: 231-251 (2018) should be added to reference 58, since Poot, 2018 extensively discusses DNA repair defetcs in relation to chromothripsis.

Revision and my comments

The reference was added as 3.

 Comment 10

Line 529: here the authors should also include: Pellestor F, Anahory T, Lefort G, Puechberty J, Liehr T, et al: Complex chromosomal rearrangements:origin and meiotic behavior.Hum Reprod Update 17: 476–494 (2011), since this reference for the first time shows that the majority of cases of complex chromotosme rearrangements are without phenotypic consequences. 

Revision and my comments

The reference was added as 64.

 Comment 11

The statement in line 530 “Ovarian dysfunction is associated with chromothripsis process” needs a reference.as well as the one in line 531.

Revision and my comments

The reference was added as 64.

 Comment 12

Lines 537 through 541 should also include Nazaryan L, Stefanou EG, Hansen C, Kosyakova

N, Bak M, et al: The strength of combined cytogenetic and mate-pair sequencing techniques

illustrated by a germline chromothripsis rearrangement involving FOXP2 . Eur J Hum Genet 22: 338–343 (2014), who were the first to empasize the need for several complementary methods to unambiguously detect chromothripsis.

 Revision and my comments

The reference was added as 64.

Comment 13

The statements in lines 545 through 554 need some elaboration. How do the authors infer chromothripsis from micronucleus formation (or abortive apoptosis) and gene amplification?

Revision and my comments

We removed this fragment because it is not subject this section.

Comment 14

In this section the authors should also discuss: Marcozzi A, Pellestor F, Kloosterman WP. The origin and characteristics of chromothripsis, Methods in Molecular Biology 1769: ??-?? (2018) and Yang J, Liu J, Ouyang L, Chen Y, Liu B, Cai H. CTLPScanner: a web server for chromothripsis-like pattern detection. Nucleic Acids Res 44:W252-W258 (2016). 

Revision and my comments

`We added information into this section.

Minor point

 Comment 15

As a rule, authors who are not born and educated in an English-speaking country should seek advice of a native speaker of English. The correct manuscript is a case in point. The authors may unknowingly have introduced statements that are difficult to grasp. Just one example: in the title the authors state “lying under”, which in English has only the literate meaning of being physically underneath. The authors may have meant the figurative meaning of “being a cause, giving rise to ...” etc. Also, the use of articles (the, a) is unusual. Therefore, this manuscript needs thorough desk editing.

We change the title and correct the English.

Best regards

Monika Lejman

Round 2

Reviewer 2 Report

Well done!